# Smooth Fractional Order Sliding Mode Controller for Spherical Robots with Input Saturation

**Ting Zhou, Yu-gong Xu and Bin Wu \***

School of Mechanical, Electronic and Control Engineering, Beijing Jiaotong University, Beijing 100044, China; 14116373@bjtu.edu.cn (T.Z.); ygxu@bjtu.edu.cn (Y.-g.X.)

**\*** Correspondence: bwu@bjtu.edu.cn; Tel.: +86-138-1171-6598

**Abstract:** This study considers the control of spherical robot linear motion under input saturation. A fractional sliding mode controller that combines fractional order calculus and the hierarchical sliding mode control method is proposed for the spherical robot. Employing this controller, an auxiliary system in which a filter was used to gain smooth control performance was designed to overcome the input saturation. Based on the Lyapunov stability theorem, the closed-loop system was globally stable and the desired state was achieved using the fractional sliding mode controller. The advantages of the proposed controller are illustrated by comparing the simulation results from the fractional order sliding mode controllers and the integer order controller.

**Keywords:** fractional sliding mode control; spherical robot; linear motion; input saturation

---

## 1. Introduction

Spherical robots are a new type of robot with a ball-shaped exterior shell. Their advantages, such as the lower energy consumption and enhanced locomotion compared to traditional wheeled or legged mobile robots have motivated many researchers to develop them with different structures in recent decades [1–6]. The inner parts are hermetically sealed inside the spherical shell, resulting in increased reliability in hostile environments, making them suitable for security and exploratory tasks [7,8].

Among the current structures of spherical robots, the pendulum-driven type is popular in industry and academia owing to its easy implementation and strong driving torque. Linear motion is the most important motion mode for the execution of a task.

However, a pendulum-driven spherical robot in a state of linear movement has underactuated and strong nonlinearity, which is similar to a class of underactuated systems such as the ball-beam system and the pendulum system. In order to overcome these problems in spherical robots or similar underacted systems, many control methods have been implemented such as adaptive control, feedback linearization, back-stepping, sliding mode control, trajectory planning, and the intelligent control method [9–19].

In all these methods, the sliding mode control approach is suitable for motion control systems owing to its unparalleled merits, such as fast response, robustness to parametric uncertainties, and resistance to disturbances [20].

To cope with the control problem of this underactuated system, the hierarchical sliding mode control method was developed [21]. This method designs the first layer sliding surface for each subsystem and then achieves the second surface by a linear or other combination of all first-layer sliding surfaces. This method demonstrates excellent convenience in the design of sliding mode controllers for underactuated systems, and it ensures the stability of the entire system, as well as of each subsystem [22,23]. Yue et al. proposed a hierarchical sliding mode controller for the velocity control of a pendulum-driven spherical robot in [24,25]. However, the traditional integral sliding surface requires

a long time to track the desired velocity and has a significant overshoot. It is worth noting that none of this research on spherical robot control systems considered input saturation. The driving capacity is relatively limited in actual spherical robots, and input saturation will be an inevitable phenomenon. Without considering the input saturation in the controller design, the control performance of a spherical robot may be weak or instability may occur in the whole system. The above research also neglects the movement of the shell in the opposite direction at the start-up stage due to the reaction force acting on the robot. As can be seen from aforementioned works, this phenomenon is caused by the conservation of momentum. Although this situation may be improved when values of control targets are higher, the problem cannot be wholly solved.

In recent years, fractional order calculus has been widely used in science and engineering. Many fractional-order controllers have been developed to enhance control performance [26–30]. For example, Podlubny proposed a fractional $PI^\lambda D^\mu$ controller in [26]. Yin et al. presented a fractional sliding mode controller and showed its benefits over integral controllers [27]. H. Delavari et al. developed a fuzzy fractional sliding surface for a nonlinear system and optimized the controller parameters using the genetic algorithm, achieving better performance than with integral sliding mode controllers [28]. Ebrahimkhani S. et al. put forward a fractional integral sliding mode controller for a wind power generation system to stabilize and enhance the robustness [29]. Zhang et al. studied a fractional sliding mode controller for a permanent magnet synchronous motor [30], while Rahmani M. et al. combined the neural network and fractional order sliding mode control and presented a peristaltic motion controller for a bionic peristaltic robot [31]. Kumar G. et al. studied a fractional sliding control with a $PI^\lambda$-$D^\mu$ sliding surface for a two-tank hybrid system and achieved better performance than that of a controller with a $PD^\mu$ sliding surface [32]. In [33], a fractional $PI^\lambda D^\lambda$ sliding controller is proposed; the PSO algorithm was used to tune the controller parameters. Furthermore, the benefits of a fractional $PI^\lambda D^\lambda$ sliding surface are shown in comparison the $PD^\mu$ sliding surface in [34]. Zhong et al. combined fractional calculus and second-order sliding mode control to give a fractional $PDD^\mu$ sliding controller for a class of nonlinear systems [35]. Aghababa M.P. et al. proposed a fractional order sliding mode controller for the finite-time stabilization of a nonautonomous fractional order underactuated system with model uncertainties and external noise [36]. Narayan et al. focused on the finite-time convergence of a fractional order nonholonomic chained system and proposed a fractional sliding order controller [37]. Based on the fractional model of a flexible underactuated manipulator, Mujumdar et al. presented a fractional $PI^\lambda$ sliding mode controller [38]. To summarize, most literature has focused on the application of fractional-order sliding mode controllers in fractional-order underactuated systems, and only very few papers have considered applying fractional sliding mode control to integer-order underactuated systems [39].

Existing papers show that the fractional sliding surfaces have better control effects than integer ones. The fractional sliding mode control has faster response and convergence speed in the initial stage due to the fractional operator. This means that the fractional sliding mode controller will yield a high output, which can more easily lead to input saturation. Moreover, in spherical robots, the output of the actuator remains constant, and when the robot system is under input saturation, it can yield a wave motion in the dynamic response process. Though many methods regarding input saturation have been developed [40–43], the wave motion in the dynamic response process cannot be solved in a satisfying way, since the method cannot completely avoid input saturation. For this reason, it is not enough to improve the spherical robot speed control under input saturation by simply designing a sliding surface with faster response and convergence speed. The opposite direction at the start-up stage and wave motion in the dynamic response process also need to be taken into consideration. Focusing on this topic, the novelty of this paper can be summarized as follows:

(1) A fractional $PI^\lambda D^\mu$ sliding control method based hierarchical sliding control and fractional calculus is proposed to improve the control performance;

(2) A novel fractional $PI^\lambda D^\mu$ sliding controller with an auxiliary system is proposed to deal with input saturation;

(3)    Smooth dynamic response is achieved by adding a filter, which can decrease the output of the controller in the initial stage and make full use of the fractional sliding surface.

In this paper, $R^+$ and $Z$ are the set of positive real numbers and integers, respectively, $R^n$ and $R^{n \times n}$ are the $n$-dimensional Euclidean space, and the matrices of order $n$. $\|.\|_\gamma$ is the norm. For a real number $x$, $\lceil x \rceil = \min\{n \in Z | x \le n\}$ is the ceiling function, and $\Gamma(x) = \int_0^\infty t^{x-1} e^{-t} dt$ is the Gamma function. $A^T$ is the transpose of matrix $A$, eig($A$) is the matrix eigenvalues.

## 2. Motion Equation and Control System

When a spherical robot is moving in linear motion mode, the dynamical model can be simplified to a two-dimensional ring-pendulum system. Figure 1 shows a diagram of the related simplified 2D model.

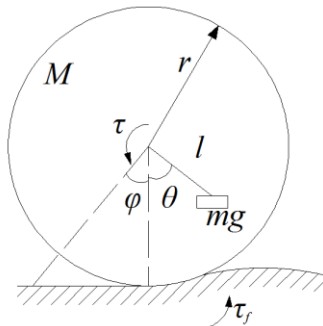

**Figure 1.** Diagram of Spherical Robot in Linear Motion.

Where $r$ is radius of sphere, $M$ is weight of shell, $m$ is weight of inner suspension, $l$ is the distance from weight center of mass to sphere center, $\varphi$ is rolling angle of the spherical robot, $\theta$ is swinging-up angle of the inner suspension relative to the shell of the spherical robot, g is the acceleration of gravity, $\tau$ is torque of motor, and $\tau_f$ is rolling friction between the shell and ground. By ignoring the damping of the actuator and the relative sliding between the spherical robot and the ground, the dynamic equation of the linear motion can be written as [44]:

$$\begin{bmatrix} (5M+3m)r^2/3 & mrl\cos\theta \\ mrl\cos\theta & ml^2 \end{bmatrix} \begin{bmatrix} \ddot{\varphi} \\ \ddot{\theta} \end{bmatrix} + \begin{bmatrix} 0 & -mrl\dot{\theta}\cos\theta \\ 0 & 0 \end{bmatrix} \begin{bmatrix} \dot{\varphi} \\ \dot{\theta} \end{bmatrix} + \begin{bmatrix} \tau_f \\ mgl\sin\theta \end{bmatrix} = \begin{bmatrix} \tau \\ \tau \end{bmatrix} \tag{1}$$

To make the controller method more convenient, the state variables are redefined as

$$x = [x_1 \quad x_2 \quad x_3 \quad x_4]^T = [\varphi \quad \dot{\varphi} \quad \theta \quad \dot{\theta}]^T \tag{2}$$

Therefore, Equation (1) transforms into the following state–space equation:

$$\begin{cases} \dot{x}_1 = x_2 \\ \dot{x}_2 = f_1(x) + b_1(x)\text{sat}(\tau) + g_1(x)\tau_f \\ \dot{x}_3 = x_4 \\ \dot{x}_4 = f_2(x) + b_2(x)\text{sat}(\tau) + g_2(x)\tau_f \end{cases} \tag{3}$$

The details of the nonlinear functions in Equation (3) are as follows

$$\begin{cases} f_1(x) = \frac{bcx_4^2 \sin x_3 + bd \sin x_3 \cos x_3}{ac - b^2 \cos^2 x_3} \\ b_1(x) = \frac{c - b\cos x_3}{ac - b^2 \cos^2 x_3} \\ g_1(x) = \frac{-c}{ac - b^2 \cos^2 x_3} \end{cases} \tag{4}$$

$$\begin{cases} f_2(x) = \frac{-ad\sin x_3 + bx_4{}^2 \sin x_3 \cos x_3}{ac - b^2 \cos^2 x_3} \\ b_2(x) = \frac{a - b\cos x_3}{ac - b^2 \cos^2 x_3} \\ g_2(x) = \frac{b\cos x_3}{ac - b^2 \cos^2 x_3} \end{cases} \tag{5}$$

The input $\mathrm{sat}(\tau)$ is defined as

$$\mathrm{sat}(\tau) = \begin{cases} \tau_{\max}, & \tau > \tau_{\max} \\ \tau, & \tau_{\min} \geq \tau \geq \tau_{\max} \\ \tau_{\min}, & \tau < \tau_{\min} \end{cases} \tag{6}$$

where $a = (5M + 3m)\, r^2/3$, $b = mrl$, $l = ml$, and $d = mgl$.

The overall system can be regarded as the shell subsystem and pendulum subsystem. Because of the existence of input saturation, there is a difference value, $\Delta\tau$, between the design input $\tau$ and the actual control input $\mathrm{sat}(\tau)$:

$$\Delta\tau = \mathrm{sat}(\tau) - \tau \tag{7}$$

The objective is to design a fractional order hierarchical sliding mode controller to control the state $x_2$ to track the desired state. With the help of this controller, better control performance and increased robustness under input saturation are achieved.

## 3. Preliminaries

This section presents some basic definitions and results of fractional calculus, which will be used later.

Among three common ways of defining fractional calculus, i.e., Grunwald-Letnikov, Riemann-Liouville, and Caputo [45], this paper adopts the Riemann-Liouville definition. The Riemann-Liouville fractional integral of a continuous function $f(t)$ is defined as:

$$_{t_0}D_t^{-p}f(t) = \frac{1}{\Gamma(p)}\int_{t_0}^{t}\frac{f(\sigma)}{(t-\sigma)^{1-p}}d\sigma, \quad (t > t_0, p > 0) \tag{8}$$

Also, the Riemann-Liouville fractional derivative of the continuous function $f(t)$ is defined as:

$$_{t_0}D_t^{p}f(t) = \frac{1}{\Gamma(q-p)}\frac{d^q}{dt^q}\int_{t_0}^{t}\frac{f(\sigma)}{(t-\tau)^{p-q+1}}d\sigma, \quad q-1 < p < q \tag{9}$$

where $p \in R^+$ is the derivative order or integral order, $q = \lceil p \rceil \in Z$; and $t_0$ is the initial value.

**Property 1 ([45]):** *For Riemann-Liouville fractional calculus, if function f(t) is continuous and $p > q > 0$, then*

$$_{t_0}D_t^{p}[_{t_0}D_t^{-q}f(t)] = {_{t_0}}D_t^{p-q}f(t), \quad t > t_0 \tag{10}$$

**Property 2 ([46]):** *If $0 < q < 1$, the following equality holds for Riemann-Liouville fractional calculus*

$$_{t_0}D_t^{q}[_{t_0}D_t^{-q}f(t)] = {_{t_0}}D_t^{-q}[_{t_0}D_t^{q}f(t)] = f(t) \tag{11}$$

**Lemma 1 ([47]):** *The fractional integration operator $D^{-q}$ with $q > 0$ is bounded*

$$\|D^{-q}f(x)\| \leq K\|f(x)\|_\gamma, \quad 1 \leq \gamma \leq \infty \tag{12}$$

*where K is a positive constant.*

**Lemma 2 ([48]):** *Consider a fractional linear time-invariant system with a different differential order as follows:*

$$D^{q_i}x(t) = Ax(t) \tag{13}$$

*where $x \in R^n$ is the state vector, $qi = [q1 \ q2 \ \dots \ qn] \in Rn, 0 < q \ i < 1$ and $A \in R^{n \times n}$ is a constant matrix. The system is stable if the following condition is satisfied*

$$\left| \arg(\text{eig}(A)) \right| > \frac{1}{2\kappa}\pi \tag{14}$$

*where the parameter $\kappa$ is the lowest common multiple of $q_i$.*

**Assumption 1.** *The state $x_i$ in Equation (3) is assumed to satisfy the following inequality*

$$\|D^p(x_i)\| \le \delta, i = 1, 2, 3, 4 \tag{15}$$

*where $\delta$ is a positive constant.*

## 4. Fractional Order Hierarchical Sliding Mode Controller

### 4.1. Design of the Fractional Hierarchical Sliding Mode Controller

The hierarchical sliding mode control method is usually used to design the controller for underactuated systems. Underactuated systems consist of serval subsystems. A first layer sliding surface is given for each subsystem, and then a second layer sliding surface is proposed. From Equation (3), the whole system is divided into an inner suspension system and a spherical shell system. Without considering the input saturation, the desired state of the system can be assumed to be:

$$x_d = [x_{1d} \quad x_{2d} \quad x_{3d} \quad x_{4d}]^T \tag{16}$$

Therefore, the tracking error variables are written as

$$e_1 = x_1 - x_{1d}, e_2 = x_2 - x_{2d} \\ e_3 = x_3 - x_{3d}, e_4 = x_4 - x_{4d} \tag{17}$$

Without considering input saturation, the shell and pendulum subsystem are rewritten in Equations (18) and (19) respectively.

$$\begin{cases} \dot{x}_1 = x_2 \\ \dot{x}_2 = f_1(x) + b_1(x)\tau + g_1(x)\tau_f \end{cases} \tag{18}$$

$$\begin{cases} \dot{x}_3 = x_4 \\ \dot{x}_4 = f_2(x) + b_2(x)\tau + g_2(x)\tau_f \end{cases} \tag{19}$$

Based on hierarchical sliding mode control method, the first-layer sliding surface for the shell subsystem is given as:

$$s_1 = k_1 D^{\lambda-1} e_2 + e_2 + k_2 D^{\mu} e_2 \tag{20}$$

The first-layer sliding surface for the inner suspension system is as follows:

$$s_2 = k_3 D^{\lambda-1} e_4 + e_4 \tag{21}$$

where $\lambda, \mu \in (0, 1)$, $k_1$, $k_2$, and $k_3$ are positive constants. To accelerate the response velocity, a fractional order operator is used in the two first-level sliding surfaces. The first-layer sliding surface in Equation (20) has a fractional $PI^{\lambda}D^{\mu}$ structure, while the structure of Equation (21) is $PI^{\lambda}$. According

the definition of fractional integral in Equation (9), the fractional integral operator has one more weight function compared to the integer order operator. The weight function has a larger value in the initial stage, but it decreases over time [45]. This means that fractional-order integrals have larger integration coefficients in the initial stage and smaller integration coefficients afterwards. If the fractional-order integral sliding mode surface is simply added to the integral-order integral sliding mode surface to form a fractional-order integral sliding mode surface, it will, on the one hand, have a faster response than the integer-order integral sliding mode surface in the initial stage, but on the other hand, may cause greater overshoot. For this reason, the first-layer sliding surface adds a fractional differential term to reduce the overshoot and further improve the response speed. The integer differential term is sensitive to errors, a feature which makes it possible to reduce the overshoot and accelerate the response speed. This also will reduce the robustness of the system. Following *Property 2*, the fractional differential term can be treated as a fractional integral of the integer order differential term. This means that the robustness of the control system can be improved by adjusting the value of fractional differential order $\mu$.

By differentiating the first layer sliding surfaces (20) and (21) with respect to time, and letting $\dot{s}_1 = 0$ and $\dot{s}_2 = 0$, then:

$$k_{11}D^\lambda e_2 + k_{12}D^{\mu+1}e_2 + f_1 + b_1\tau + g_1\tau_f - \dot{x}_{2d} = 0 \tag{22}$$

$$k_{12}D^\lambda e_2 + f_2 + b_2\tau + g_2\tau_f - \dot{x}_{4d} = 0 \tag{23}$$

Then, the equivalent control law of each subsystem is as follows:

$$\tau_1 = -b_1^{-1}(k_{11}D^\lambda e_2 + \lambda_{12}D^{\mu+1}e_2 + f_1 + g_1\tau_f - \dot{x}_{2d}) \tag{24}$$

$$\tau_2 = -b_2^{-1}(k_2D^\lambda e_4 + \dot{x}_{4d} - f_2 - g_2\tau_f) \tag{25}$$

The total control law must include some portions of the equivalent control law of each subsystem to guarantee that the first sliding simultaneously converges to zero. Furthermore, the total control law also needs to include some portions of switch control law. Then, the total control law $\tau$ is defined as

$$\tau = \tau_1 + \tau_2 + \tau_{sw} \tag{26}$$

where $\tau_{sw}$ represents the switch law.

To achieve the switch law, the second layer sliding surface is designed as follows:

$$S = \eta s_1 + \xi s_2 \tag{27}$$

where $\eta$ and $\xi$ are positive constants. From variable structure theory, the exponential reaching rate is selected to obtain better dynamic performance. By calculating the differentiation of the second sliding surface (27),

$$\begin{aligned} \dot{S} &= \eta(k_{11}D^\lambda e_2 + k_{12}D^{\mu+1}e_2 + f_1 + b_1\tau + g_1\tau_f - \dot{x}_{2d}) + \\ &\quad \xi(k_{12}D^\lambda e_2 + f_2 + b_2\tau + g_2\tau_f - \dot{x}_{4d}) \\ &= -\alpha\text{sign}(S) - \beta S \end{aligned} \tag{28}$$

where $\alpha$ and $\beta$ are positive constants.

Considering the differentiation of the second layer sliding surface (27), the switch control law is given as

$$\tau_{sw} = \frac{-\alpha\text{sign}(S) - \beta S + \eta b_2\tau_1 + \xi b_1\tau_2}{\eta b_1 + \xi b_2} \tag{29}$$

Placing Equations (24), (25), and (29) into Equation (26), the total control law is given as follows:

$$\tau = \frac{-\alpha \text{sign}(S) - \beta S + \eta b_1 \tau_1 + \xi b_2 \tau_2}{\eta b_1 + \xi b_2} \tag{30}$$

### 4.2. A Novel Fractional $PI^\lambda D^\mu$ Sliding Mode Controller

Equation (18) uses the fractional differential term in the sliding surface; this term leads to faster response and lower overshoot. However, it may lead to a high controller output when the desired object varies in impulse form due to the explosion of the derivation pertaining to the desired speed, $x_{2d}$. The large output of the controller will cause the actuator to saturate rapidly. It also makes the shell subsystem tracking the desired speed become wave-like, with an opposite speed response at the start stage. Therefore, inspired by backstepping method m [49], this study applies a filter to resolve this problem. Below, the aforementioned results are extended to deal with the input saturation problems of the spherical robot system and to achieve smoother dynamic response.

Let $x_{2d}$ pass the filter $\bar{x}_{2d}$. Then, the new desired state is redefined in Equation (31)

$$\varepsilon \dot{\bar{x}}_{2d} + \bar{x}_{2d} = x_{2d} \tag{31}$$

where $\varepsilon$ is a designed positive constant, and the initial value of the filter is $\bar{x}_{2d} = 0$. Then, the error $e_2$ may be redefined as:

$$e_2 = x_2 - \bar{x}_{2d} \tag{32}$$

The new equivalent control law of each subsystem is updated as:

$$\hat{\tau}_1 = -b_1^{-1}(\lambda_{11}D^\lambda e_2 + \lambda_{12}D^{\mu+1}e_2 + f_1 + g_1\tau_f - \dot{\bar{x}}_{2d}) \tag{33}$$

$$\hat{\tau}_2 = -b_2^{-1}(\lambda_2 D^\lambda e_4 + \dot{x}_{4d} - f_2 - g_2\tau_f) \tag{34}$$

In order to handle the input saturation, an auxiliary system was designed. The total control law needs to include some terms of the auxiliary system. Hence, the new total control law $\hat{\tau}$ can be updated as:

$$\hat{\tau} = \frac{-\alpha \text{sign}(S) - \beta S + \eta b_1 \hat{\tau}_1 + \xi b_2 \hat{\tau}_2 - \hat{S}}{\eta b_1 + \xi b_2} \tag{35}$$

$$\dot{\hat{S}} = \begin{cases} -\left\{[S(\eta b_1 + \xi b_2)\Delta\tau + \frac{1}{2}\Delta\tau^2]/\hat{S}\right\} - \chi\hat{S} + \Delta\tau, & |\hat{S}| \geq \rho \\ 0, & |\hat{S}| < \rho \end{cases} \tag{36}$$

where $\chi$ is a positive constant, and $\rho$ is a positive constant whose value is small.

### 4.3. Stability Analysis of Each Surface

**Theorem 1.** *For the dynamic system of a spherical robot which satisfies the constraint of Equation (14), and the control parameters $\alpha$, $\beta$, and $\chi$ satisfy $\alpha > 1$, $\beta > 0$, $\chi > 1/2$, the first-level sliding surface described in Equations (20) and (21) are asymptotically stable, and the second sliding surface Equation (27) is uniformly bounded by the control laws defined in Equations (35) and (36).*

**Proof.** Choose the Lyapunov function as

$$V_1 = \frac{1}{2}S^2 + \frac{1}{2}\hat{S}^2 \tag{37}$$

□

The derivative of Equation (37) as

$$
\begin{aligned}
\dot{V}_1 &= S\dot{S} + \hat{S}\dot{\hat{S}} \\
&= S[\eta(k_1 D^\lambda e_2 + k_2 D^{\mu+1} e_2 + f_1 + b_1(\tau + \Delta\tau) + g_1\tau_f - \dot{\bar{x}}_{2d}) + \\
&\quad \xi(k_3 D^\lambda e_4 + f_2 + b_2(\tau + \Delta\tau) + g_2\tau_f - \dot{x}_{4d})] + \hat{S}\dot{\hat{S}} \\
&= S[-\eta b_1\tau_1 - \xi b_2\tau_2 + (\eta b_1 + \xi b_2)\tau + (\eta b_1 + \xi b_2)\Delta\tau] + \hat{S}\dot{\hat{S}}
\end{aligned}
\tag{38}
$$

Equation (38) needs to take the following two cases into account.

**Case 1.** *If $\left|\hat{S}\right| \geq \rho$, by putting Equation (36) into Equation (38), one can obtain*

$$
\begin{aligned}
\dot{V}_1 &= S\dot{S} + \hat{S}\dot{\hat{S}} \\
&= S[\eta(k_1 D^\lambda e_2 + k_2 D^{\mu+1} e_2 + f_1 + b_1(\hat{\tau} + \Delta\tau) + g_1\tau_f - \dot{\bar{x}}_{2d}) + \\
&\quad \xi(k_3 D^\lambda e_4 + f_2 + b_2(\hat{\tau} + \Delta\tau) + g_2\tau_f - \dot{x}_{4d})] + \hat{S}\dot{\hat{S}} \\
&= S[-\eta b_1\hat{\tau}_1 - \xi b_2\hat{\tau}_2 + (\eta b_1 + \xi b_2)\hat{\tau} + (\eta b_1 + \xi b_2)\Delta\tau] - \\
&\quad S(\eta b_1 + \xi b_2)\Delta\tau - \tfrac{1}{2}\Delta\tau^2 - k_3\hat{S}^2 + \hat{S}\Delta\tau \\
&\leq -\alpha S^2 - \beta|S| - \tfrac{1}{2}\Delta\tau^2 - \chi\tfrac{1}{2}\hat{S}^2 + \tfrac{1}{2}\hat{S}^2 + \tfrac{1}{2}\Delta\tau^2 \\
&\leq -\alpha S^2 - (\chi - \tfrac{1}{2})\hat{S}^2 - \beta|S| \\
&\leq -u_1 V_1
\end{aligned}
\tag{39}
$$

*where $u_1 = \min\{\alpha, \chi - \tfrac{1}{2}\}$.*

**Case 2.** *If $\left|\hat{S}\right| < \rho$, one can obtain*

$$
\begin{aligned}
\dot{V}_1 &= S\dot{S} + \hat{S}\dot{\hat{S}} \\
&= S[-\eta b_1\hat{\tau}_1 - \xi b_2\hat{\tau}_2 + (\eta b_1 + \xi b_2)\hat{\tau} + (\eta b_1 + \xi b_2)\Delta\tau + \hat{S}] \\
&= -\alpha S^2 - \beta|S| + S(\eta b_1 + \xi b_2)\Delta\tau + S\hat{S} \\
&\leq -\alpha S^2 - \beta|S| + \tfrac{1}{2}S^2 + \tfrac{1}{2}[(\eta b_1 + \xi b_2)\Delta\tau]^2 + \tfrac{1}{2}S^2 + \tfrac{1}{2}\hat{S}^2 \\
&\leq -(\alpha - 1)S^2 - \tfrac{1}{2}\hat{S}^2 + \tfrac{1}{2}[(\eta b_1 + \xi b_2)\Delta\tau]^2 \\
&\leq -u_2 V_1 + \tfrac{1}{2}[(\eta b_1 + \xi b_2)\Delta\tau]^2
\end{aligned}
\tag{40}
$$

*where $u_2 = \min\{\alpha - 1, \tfrac{1}{2}\}$.*

Thus, one can conclude that the second sliding surface $S$ is uniformly ultimately bounded.

When $\eta, \xi$ satisfy the condition $\eta, \xi > 0$, the convergence of the function is independent of the specific value of $\eta, \xi$. Considering the following two sliding surfaces:

$$
S_1 = \eta_1 s_1 + \xi s_2
\tag{41}
$$

$$
S_2 = \eta_2 s_1 + \xi s_2
\tag{42}
$$

where $\eta_1, \eta_2, \xi$ are positive constant and $\eta_1 \neq \eta_2$. Here, one supposes that $|S_1| \geq |S_2|$

$$
\begin{aligned}
\int_0^t (S_1^2 - S_2^2)d\sigma &= \int_0^t [\eta_1^2 s_1^2 - \eta_2^2 s_1^2 + 2\xi s_1 s_2(\eta_1 - \eta_2)]d\sigma \\
&= \int_0^t [-(\eta_1 - \eta_2)^2 s_1^2 + 2(\eta_1 - \eta_2)s_1 S_1]d\sigma > 0
\end{aligned}
\tag{43}
$$

From Equation (43)

$$
\int_0^t (\eta_1 - \eta_2)^2 s_1^2 d < \int_0^t 2(\eta_1 - \eta_2)s_1 S_1 d\sigma
\tag{44}
$$

Then

$$\int_0^t s_1^2 d\sigma < \int_0^t \frac{2s_1 S_1}{(\eta_1 - \eta_2)} d\sigma < \int_0^t \frac{2\|s_1\|_\infty \|S_1\|_\infty}{(\eta_1 - \eta_2)} d\sigma \tag{45}$$

The actuator directly drives the inner suspension, so one can see that $x_3$ and $\dot{x}_4$ are bounded. By Lemma 1, $\lambda_2 D^{\alpha-1} e_4 < \infty$ can be obtained. Therefore, the fact that sliding surface $s_2$ is bounded can be confirmed. Then, according to Assumption 1, one can obtain $\dot{s}_2 < \infty$. Based on the conclusion that $S, \dot{S} < \infty$, one obtains $s_1, \dot{s}_1 < \infty$.

Furthermore

$$\int_0^t s_1^2 d\sigma < \infty \tag{46}$$

with the Barbalat lemma [50], $\lim_{t\to\infty} s_1 = 0$; similarly, $\lim_{t\to\infty} s_2 = 0$ is achieved.

**Theorem 2.** *If $k_1 > 0$, $k_2 > 0$, $k_3 > 0$, $\lambda, \mu \in (0,1)$, and the constraint Equation (14) is satisfied, then the error of the two sliding surfaces in Equations (20) and (21) decays asymptotically toward zero.*

**Proof.** When the first-layer sliding surfaces remain on the respective manifold, $s_1 = 0$ and $s_2 = 0$; the first sliding mode surface $s_1$ is rewritten into the state equation form as follows:

$$\begin{bmatrix} D^{1-\lambda}(D^{\lambda-1}(e_2)) \\ D^\mu(e_2) \end{bmatrix} = \begin{bmatrix} 0 & 1 \\ -k_1/k_2 & -1/k_2 \end{bmatrix} \begin{bmatrix} D^{1-\lambda}(e_2) \\ e_2 \end{bmatrix} \tag{47}$$

□

By redefining the vector $Y = [y1 \ y2]T = [D1 - \lambda(e2) \ e2]T$, Equation (47) is rewritten as:

$$\begin{bmatrix} D^{1-\lambda}(y_1) \\ D^\mu(y_2) \end{bmatrix} = \begin{bmatrix} 0 & 1 \\ -k_1/k_2 & -1/k_2 \end{bmatrix} \begin{bmatrix} y_1 \\ y_2 \end{bmatrix} \tag{48}$$

Equation (48) becomes:

$$\det \begin{pmatrix} s^{1-\lambda} & -1 \\ -k_1/k_2 & s^\mu + 1/k_2 \end{pmatrix} = 0 \tag{49}$$

Then

$$k_2 s^{1-\lambda+\mu} + k_2 s^{1-\lambda} + k_1 = 0 \tag{50}$$

Assuming that $s = j\omega = |\omega|(\cos\frac{\pi}{2} + j\sin\frac{\pi}{2})$ is a root of Equation (50), one gets

$$k_2|\omega|^{1+\mu-\lambda}(\cos\frac{(1+\mu-\lambda)\pi}{2} + j\sin(\pm\frac{(1+\mu-\lambda)\pi}{2})) + \\ k_2|\omega|^{1-\lambda}(\cos\frac{(1-\lambda)\pi}{2} + j\sin(\pm\frac{(1-\lambda)\pi}{2})) + k_1 = 0 \tag{51}$$

Separating the real and imaginary parts of Equation (51)

$$k_2|\omega|^{1+\mu-\lambda}\cos\frac{(1+\mu-\lambda)\pi}{2} + k_2|\omega|^{1-\alpha}\cos\frac{(1-\lambda)\pi}{2} + k_1 = 0 \tag{52}$$

$$k_2\sin(\pm\frac{(1+\mu-\lambda)\pi}{2}) + k_2\sin(\pm\frac{(1-\lambda)\pi}{2}) = 0 \tag{53}$$

Taking the sum of squares of Equations (52) and (53)

$$
\begin{aligned}
& k_2^2|\omega|^{2(1+\mu-\lambda)} + k_2^2|\omega|^{2-2\lambda} + k_1^2 + 2k_2^2\cos\frac{\mu\pi}{2} + \\
& 2k_1k_2\left(\cos\frac{(1+\mu-\lambda)\pi}{2} + \cos\frac{(1-\lambda)\pi}{2}\right) = 0
\end{aligned}
\tag{54}
$$

Because of the parameters to hold the conditions that $k_1 > 0$, $k_2 > 0$ and $\lambda, \mu \in (0,1)$, Equation (54) has no real solutions. This means that Equation (54) has no purely imaginary roots. Combining the constraint of Equation (14), one determines that the errors in the sliding surface can decay asymptotically toward zero. Similarly, the first sliding mode surfaces $s_2$ is rewritten as

$$
D^{1-\lambda}e_4 = -\frac{1}{k_3}e_4
\tag{55}
$$

The eigenvalue of Equation (55) satisfies $\left|\arg(\mathrm{eig}(-\frac{1}{k_3}))\right| = \pi > \frac{1-\lambda}{2}\pi$. So, one may conclude that state $e_4$ can converge asymptotically to zero.

**Remark 1.** *Chattering phenomena occur when the control law $\tau$ is applied to the system. Many approaches have been developed to reduce the chattering caused by the switch function, such as the robust adaptive approach, the higher-order sliding mode method, and the application of continuous or other switch functions [51–53]. This study chooses the hyperbolic tangent function instead of the sign function in the switching law to weaken chattering. Accordingly, the total control law Equation (28) can be written as:*

$$
\tau = \frac{\alpha\tanh(S) + \beta S - \eta b_1\tau_1 - \xi b_2\tau_2 - \hat{S}}{\eta b_1 + \xi b_2}
\tag{56}
$$

where

$$
\tanh(S) = \frac{e^s - e^{-s}}{e^s + e^{-s}}
\tag{57}
$$

**Remark 2.** *A selection range of controller parameters is given in Theorems 1 and 2, although a method by which to choose all parameters to gain better control performance still needs to be discussed. Hence, the following steps to adjust the controller parameters are based on simulations.*

Step 1: Set $\lambda = \mu = 0$, $k_2 = 0$, and $\varepsilon = 0$ to make the first layer sliding surface the same as an integer order integral sliding surface. Then, select a large enough value of $\alpha$ and $\beta$ in the switch law $\tau_{sw}$ to achieve good robustness and a high convergence rate.

Step 2: Select the same value for the parameters in the second layer surface. Because the first layer sliding surface should satisfy the condition that $s_1 = 0$ and $s_2 = 0$ at the same time. It is better to choose the same value for the parameters of the second layer surface.

Step 3: Select the initial value of $k_1$ and $k_3$. It is better to make $k_3$ larger than $k_1$. Because the pendulum subsystem is the direct actuator for the shell subsystem. The parameter $k_3$ is the coefficient of the integral term in the sliding surface $s_2$ of the pendulum subsystem, a fast response rate can be achieved by increasing the value of it.

Step 4: Increase the value of fractional integral order $\lambda$ to gain a faster responding rate, then increase the value of $\mu$ to shorten the adjustment time and reduce the overshoot.

Step 5: Increase the value of $\varepsilon$ to smooth the dynamic response of the spherical robot. A higher value of $\varepsilon$ will cause a longer adjustment time.

## 5. Simulation Study

This section presents the results of numerical simulations in the MATLAB software, which were performed to demonstrate the effectiveness of the proposed novel fractional $PI^\lambda D^\mu$ sliding mode controller (NFO-PID SMC). Table 1 lists the main physical parameters of the spherical robot. To test

the performance of the controller, the desired angular velocity of the spherical robot was set to $x_{2d} = 10$ rad/s. State $x_1$ is the rolling angle of the spherical robot; thus, the desired state $x_{1d}$ was set to $x_{1d} = 10t$ rad. The shell system was driven by the eccentric moment produced by the rotation between the shell and pendulum subsystems. When the shell keeps moving at a constant velocity, the swinging-up angular velocity of the inner suspension relative to the shell should be zero [20]. This way, the desired state vector of the control system of Equation (3) can be determined.

$$x_d = [10t \quad \bar{x}_{2d} \quad 0 \quad 0] \tag{58}$$

**Table 1.** Main physical parameters of the spherical robot.

| Parameter | $M$/kg | $m$/kg | $r$/m | $l$/m | $\tau_{max}$/N·m | $\tau_{min}$/N·m |
|:---:|:---:|:---:|:---:|:---:|:---:|:---:|
| **Value** | 2.5 | 8 | 0.15 | 0.09 | 2.3 | −2.3 |

Yue et al. [19] proposed an integer-type order sliding mode surface for spherical robot speed control. For comparison, an integer-order sliding controller (IO-SMC) based on the method proposed in this paper was implemented for the spherical robot system. The integer-type order sliding mode surface equation can be written as follows:

$$s_1 = k_1 e_1 + e_2 \tag{59}$$

$$s_2 = k_3 e_3 + e_4 \tag{60}$$

Table 2 summarizes the parameters of the fractional-order hierarchical sliding mode controller with the $PI^\lambda D^\mu$ sliding surface. In order to highlight the differences between the controllers, some parameters were kept the same for both controllers. Furthermore, two additional controllers without the filter were used by setting $\varepsilon = 0$. To verify the robustness of the four controllers, suppose the friction force $\tau_f = 0$ at time $t = 0$ s, and then set the friction force $\tau_f = 1$ N·m at $t = 15$ s.

**Table 2.** Parameters of the controllers.

| Parameter | $k_1$ | $k_2$ | $k_3$ | $\eta$ | $\zeta$ | $\alpha$ | $\beta$ | $\lambda$ | $\mu$ | $\varepsilon$ |
|:---:|:---:|:---:|:---:|:---:|:---:|:---:|:---:|:---:|:---:|:---:|
| **Value** | 1 | 0.3 | 12 | 2 | 2 | 5 | 5 | 0.1 | 0.1 | 1 |

Figure 2 shows the curves of the velocity response of the spherical shell, while Figure 3 shows the curves of the errors. Figure 4 shows the control output. Figures 5–7 show the time evolutions of all sliding mode surfaces. Table 3 summarizes the performance of the two controllers in terms of overshoot and adjustment time.

**Table 3.** Performances of the controllers.

| Controller | Overshoot (%) | Adjustment Time (s) |
|:---:|:---:|:---:|
| NFO-PID SMC | 0 | 3.4 |
| IO-PI SMC | 35 | 6.1 |

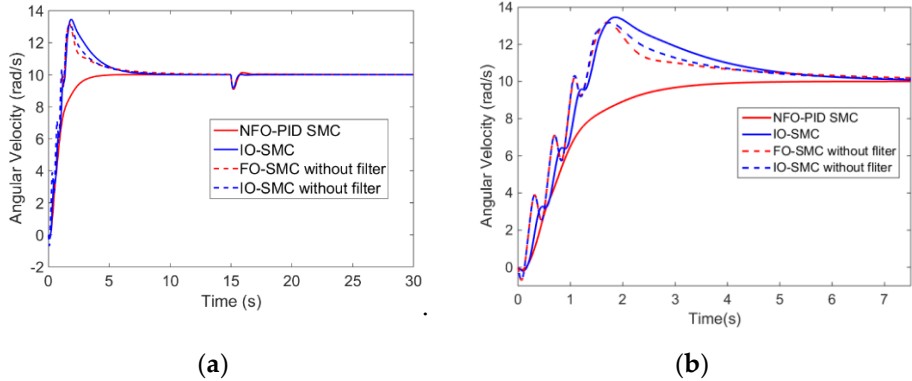

**Figure 2.** Comparison of the curves of the velocity of shell response among four controllers: (**a**). Time interval 0~30 s; (**b**). Time interval 0~7 s.

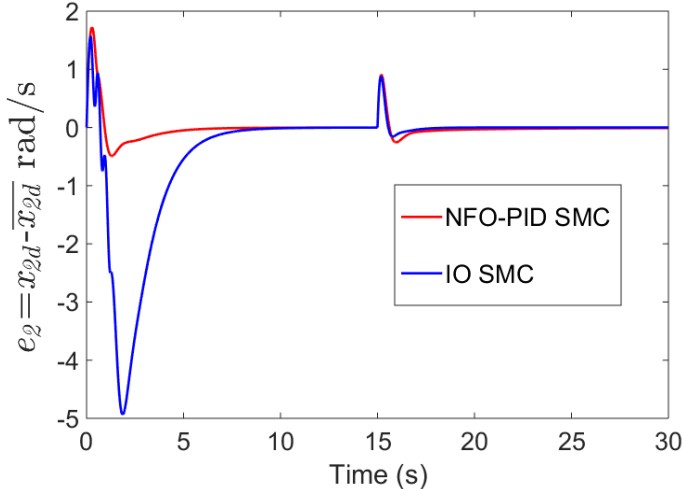

**Figure 3.** Curves of error $e_2$.

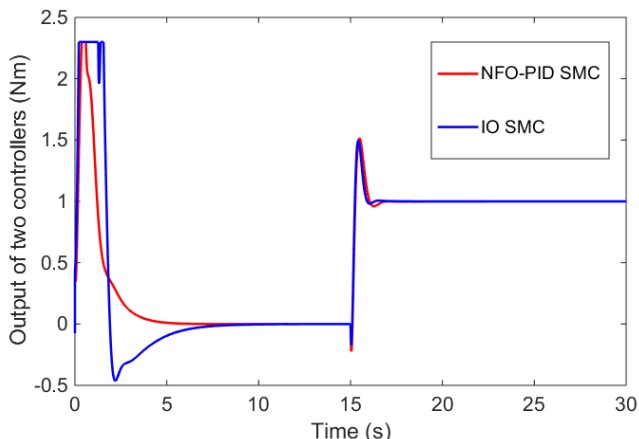

**Figure 4.** The output of two controllers.

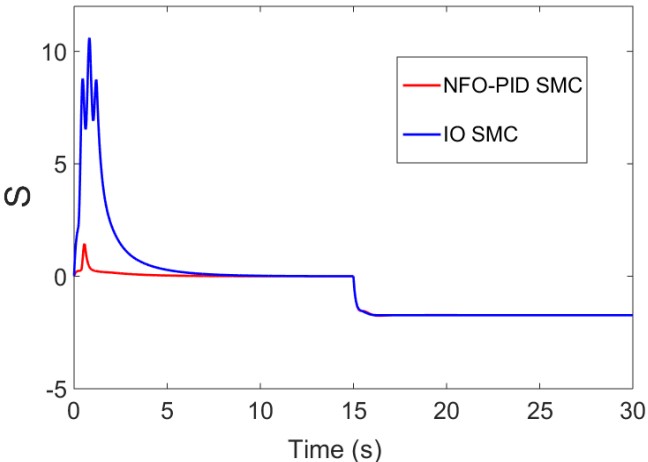

**Figure 5.** Curves of second-layer sliding mode surfaces $S$.

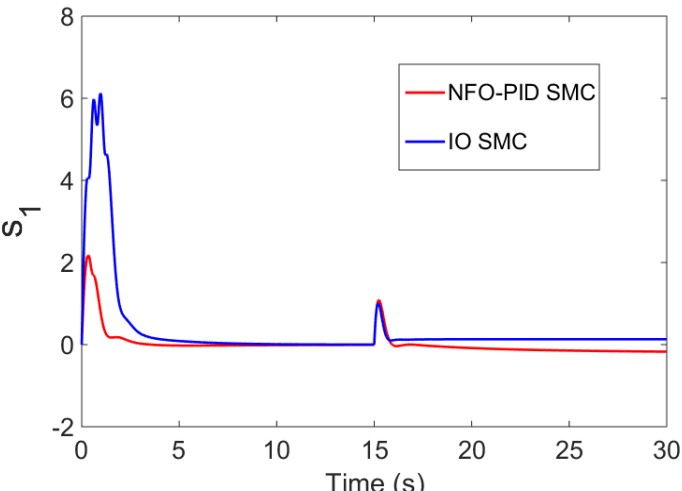

**Figure 6.** Curves of first-layer sliding mode surfaces $s_1$.

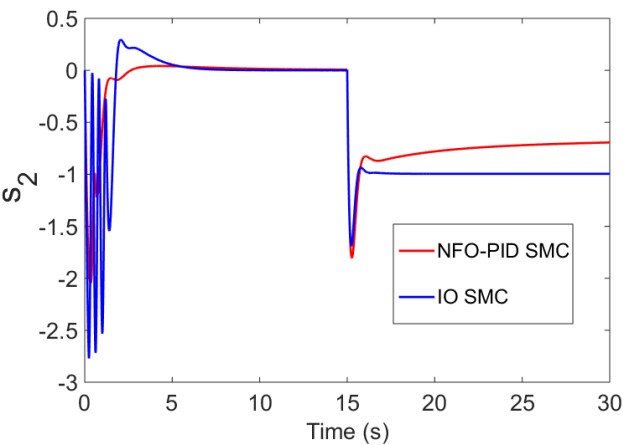

**Figure 7.** Curves of first-layer sliding mode surfaces $s_2$.

### 5.1. Tracking Performance Analysis

The results in Figure 2a indicate that all four controllers can track the desired angular velocity. The NFO-PID SMC has a shorter adjustment time and does not have overshoot. The adjustment time

is shortened from 6.1 s to 3.4 s. In Figure 2b, it can be seen that the controller without a filter causes wave-like behavior and has a larger opposite speed at the start stage. However, the proposed method yields a smoother dynamic response, showing that the filter is effective for the smooth control of spherical robots. Though the filter is used in the IO SMC, the wave-like response still occurs in the control process, but this phenomenon is hardly observed in the NFO-PID SMC. Figure 3 proves that the NFO-PID SMC and IO SMC can both track the output of the filter $\bar{x}_{2d}$; furthermore, the NFO-PID SMC controller forces error $e_2$ to begin to converge earlier than the IO SMC does. This case also shows that the NFO-PID SMC has better tracking performance than IO SMC.

Figure 4 shows that the NFO-PID SMC controller can provide the spherical robot with smooth input. The output of the two controllers reaches the maximum limit in a very short time, but the output of IO-SMC stays in the maximum value for a longer time than the NFO-PID SMC. This explains why the wave motion still occurs in the IO SMC. Figure 5 highlights the better dynamic response of the second-layer sliding surfaces of the NFO-PID SMC controller compared to the IO-SMC controller. Figures 6 and 7 show that the shell and pendulum subsystems are asymptotically stable, and the fractional sliding surface proposed in Equations (24) and (25) converges to zero faster than the integral sliding surface. This indicates that the fractional sliding mode surface proposed in this paper is effective. In summary, the NFO-PID SMC controller successfully tracks the desired velocity, and the goal of better control performance is achieved.

### 5.2. Robustness Analysis

As shown in Figure 3, the robot system actuated by the four controllers can recover to the desired velocity almost at the same time when subjected to friction interference at the time of 15 s. This shows the robustness of the sliding mode control method. The results of Figure 3 after 15 s indicate that the NFO-PID SMC has lower recovery speed compared to IO SMC. The same situation also can be observed in Figures 6 and 7. This situation, caused by the weigh function in the fractional integral operation, decreases over time. The ability of the integra term to tolerate extra disturbers is weakened. Figure 5 shows that the second-layer sliding surfaces of the two controllers reach a new steady-state which is not equal to zero when the rolling friction changes from 0 to 1 N·m. The new steady-state of each second-layer sliding surface is the same. All the results show that NFO-PID SMC has the same robustness as IO SMC, though a fractional differential term is included.

## 6. Conclusions

In this study, a novel fractional $PI^\lambda D^\mu$ sliding controller is proposed to improve the control performance of the spherical robot linear motion with input saturation. The fractional sliding surface is applied in the control design to achieve quick tracking of the control target. An auxiliary system is designed to handle input saturation. By adding a filter in the desired velocity to reduce the initial value and gain a smooth dynamic response, simulation results show that the novel fractional $PI^\lambda D^\mu$ sliding mode controller has smaller overshoot and a shorter adjustment time than the integer one. The adjustment time of spherical robot system decreases by 44%, and presents no overshoot. The wave motion in the dynamic response process is efficiently suppressed. To conclude, the presented control method can be extended to a class of underactuated systems, and further work should focus on the implementation of the proposed method in a real spherical robot to verify its effectiveness.

**Author Contributions:** Methodology, T.Z. and B.W.; Simulation; T.Z.; Writing, T.Z., Y.-g.X. and B.W. All authors have read and agree to the published version of the manuscript.

**Funding:** This research was funded by The Fundamental Research Funds for the Central Universities under Grant No.2019JBM408.

**Conflicts of Interest:** The authors declare no conflict of interest.

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
