# Peer review of "Smooth Fractional Order Sliding Mode Controller for Spherical Robots with Input Saturation"

_applsci, doi:10.3390/app10062117_

Round 1

Reviewer 1 Report

In this paper, the authors designed a fractional-order sliding mode controller (with PIλDμ and PIλ structure) for velocity control of the linear motion of a spherical robot under input saturation. They added a filter, inspired by the backstepping method, to smooth the dynamic response of the proposed controller. Finally, they used numerical simulations for validating the controller and comparing its performance with an integer-order sliding mode controller. It is concluded that the proposed method leads to a faster response and lower overshoot. In general, the paper's contribution is very marginal. Although the authors have tried to convey the main concepts, the formulations are vague and hard to follow. Moreover, the paper text and style need to be modified significantly. Here are some comments:

Major Comments:
1- Can this method be extended for the non-linear (or non-planar) motion of a spherical robot?
2- How robust the controller can be in the presence of slippage at the contact of the spherical robot with the ground?
3- What is the physical meaning of Assumption 1 given in (15)?
4- The authors should first explain the basis of the Fractional Order Hierarchical Sliding Mode Controller in more detail and general form before defining the first-layer sliding surfaces in (18) and (19) to make the formulation clearer.
5- How did you derive (20) and (21)? Are they valid only when \tau_min<= \tau <= \tau_max? If yes, this should be mentioned clearly.
6- You should explain how you derived the total control law in (23)?
7- Why did you define the new desired state given in (24) in this form? What is the physical meaning of this definition?
8- Why did you assume that the desired values for \theta and \theta_d to be zero in (52)?
9- How did you come up with the controller parameters given in Table 2?
10- In this paper, the controller performance is compared with only one more controller (i.e., an integer-order sliding mode controller). It would be better to compare the result with some more controllers.
11- Experimental results are highly recommended in order to show the capability of the proposed controller.

Minor Comments:
There are some typos and repeated sentences in the paper as follows. The authors should correct these errors and also revise the paper style.
1- The word "The" in the first paragraph of the Introduction section is repeated twice.
2- The sentence "where ε is a designed positive constant. Then the error e2 redefines as" after equation (24) is repeated twice.
3- The paragraph "Figure 2 shows the curves of velocity response of the spherical shell, while Figure. 3 shows the curves of the errors. Table 3 summarizes the performances of the two controllers in terms of overshoot and adjust time.", in the Simulation Study section, is repeated twice.
4- In the sentence "Figure 6 shows that the NFO-PID SMC controller can provide the spherical robot with a smooth input.", in the Simulation Study section, Figure 6 should be changed to Figure 4.

Reviewer 2 Report

  1.  

I have the following comments:

  1. The contribution of the paper is questionable. The motivation and background of research concerning fractional order systems (FOS) control should be more clear and explained in detail in Section 4.
  2. The proposed procedure is fully formalized. Although the paper is full of mathematical formulas, the theoretical contribution is minimal. The utility of FOS methods, in the proposed methodology, is marginal treated and the paper does not clarify the necessity of a complex control law as (23). The dynamic model is defined by Integer order Models. Why and how can you select the order alpha and beta for the FOS control law? In Simulation Study you propose a control law but you do not specify the values used for lambda and miu.
  3. The authors necessitate clarifying the following: The effectiveness of the proposed approach is not supported by a comparison with other control approach. Since the proposed system is rather complex the authors should show that it permits to gain in terms of performance with respect to a standard method.
  4. The readability of the manuscript should be improved, and the organization and structure of the paper may be adjusted.

Reviewer 3 Report

Nice paper, interesting view but paper needs serious improvement.

First notice - graphics. Not defined main terms on initial figure 1and their explanation of figure caption, missing axis legend or units in fig. 3, 5-7 as well as unclear figures captions.

Introduction need rewriting and should provide implementation of the modeled device, provide main problems and solution, offered int his particular paper.

Task for research need to state in evident form, otherwise unclear result values - did they achieve aim or not.

Nomenclature of used variables desired, as you using big expressions and it is unclear when following them.

Result section needs system and explanations.

Conclusions: rewrite or simply create it, while now it cannot be seen real value of research and noticed effects.

Reviewer 4 Report

1. The overview of publications devoted to spherical robots and control algorithms  is insufficiently complete, the more so as the authors consider control algorithms that take into account not only straight-line motion, but also different actuating mechanisms.

We recommend the authors to familiarize themselves with the following papers:

  • Bai Y., Svinin M., Yamamoto M.,  Dynamics-Based Motion Planning for a Pendulum-Actuated Spherical Rolling Robot, Regular and Chaotic Dynamics, 2018, vol. 23, no. 4,  pp. 372-388
  • Borisov A. V., Kilin A. A., Karavaev Y. L., Klekovkin A. V., Stabilization of the motion of a spherical robot using feedbacks, Applied Mathematical Modelling, 2019, vol. 69, pp. 583-592
  • Kilin A. A., Pivovarova E. N., Ivanova T. B., Spherical Robot of Combined Type: Dynamics and Control, Regular and Chaotic Dynamics, 2015, vol. 20, no. 6, pp. 716-728
  • Roozegar  M., Mahjoob M. J., Ayati M.,  Adaptive Estimation of Nonlinear Parameters of a Nonholonomic Spherical Robot Using a Modified Fuzzy-based Speed Gradient Algorithm, Regular and Chaotic Dynamics, 2017, vol. 22, no. 3,  pp. 226-238
  • Ivanova T. B., Kilin A. A., Pivovarova E. N., Controlled Motion of a Spherical Robot with Feedback. II, Journal of Dynamical and Control Systems, 2019, vol. 25, no. 1, pp. 1-16
  • Karavaev Y. L., Kilin A. A., Nonholonomic Dynamics and Control of a Spherical Robot with an Internal Omniwheel Platform: Theory and Experiments, Proceedings of the Steklov Institute of Mathematics, 2016, vol. 295, pp. 158-167
  • Ivanova T. B., Kilin A. A., Pivovarova E. N., Control of the Rolling Motion of a Spherical Robot on an Inclined Plane, Doklady Physics, 2018, vol. 63, no. 10, pp. 435-440

2. The proposed novel fractional-order sliding mode controller for the velocity of control of linear motion is compared with another, but, for ease of comparison, the same controller parameters (see Table 2) are used. It is unclear what the motivation for the choice of these values is.  Is it possible to achieve better values of overshoot       and adjust time for other parameter settings of the IO-PI SMC controller?

3. There is no explanation for the values of the physical parameters that were used for simulation. Some of them look strange, for example, the mass of the inner suspension, which is 8 kg.  It is rather difficult to place such a mass at a distance of 0.09 m when the radius of the spherical shell is 0.09 m.

The manuscript is recommended for publication after revision.

Round 2

Reviewer 1 Report

The authors have addressed all the major issues and revised the manuscript significantly. Therefore, the manuscript has reached an acceptable level and I have no further comments on the revised manuscript.

Reviewer 2 Report

The revised work satisfies the specified observations.

However, the selection technique of control parameters presented in Remark 2 is quite empirical.

Reviewer 3 Report

Acceptable changes, but English need polishing, especially in altered text area..

Reviewer 4 Report

The authors have taken into account all my comments and suggestions, I think the revised version can be published in Applied Sciences.